# Using electronic health records to assess the relationship between colonization pressure and nosocomial pathogen acquisition

Hospitalized patients are at risk for developing hospital acquired infections. Active surveillance for bacterial colonization is effective at preventing infection but is resource-intensive and limited to high-risk units and a subset of high-risk pathogens. Colonization pressure, defined as the prevalence of an organism among ward co-occupants, has been associated with hospital acquired infection and can be calculated in real-time using data in the electronic health record, but its use by infection control programs has been limited. As a proof-of-concept study, we built an open source informatics tool to model the impact of colonization pressure on nosocomial acquisition for a wide range of drug susceptible and resistant pathogens using electronic health record data from a large integrated health system in the Northeast United States, collected between May 2015 and July 2024. Using a matched case-control design, we show a consistent positive association between colonization pressure for an organism and nosocomial acquisition of that organism, regardless of whether that organism was drug susceptible or resistant. We also observed significant positive relationships between disparate organisms (e.g., colonization pressure from vancomycin resistant *Enterococcus faecium* and hospital acquisition of extended-spectrum beta-lactamase producing *Klebsiella pneumoniae*) as well as negative associations, primarily between organisms that inhabit distinct niches, such as colonization pressure from drug resistant *Pseudomonas aeruginosa* and hospital acquisition of vancomycin susceptible *Enterococcus faecalis*. Our results suggest nosocomial transmission of potential pathogens is widespread in a tertiary care hospital system with advanced infection control programs. We have made the software and our patient-level dataset publicly available to support future research and infection control interventions.

Hospital acquired infection is a common and serious complication of medical care[1]. A primary mechanism for this begins when a patient who is colonized with a potential pathogen is admitted to the hospital. That individual becomes a reservoir from which the hands and clothing of healthcare workers, hospital equipment and room surfaces are contaminated[2]. Transfer from these contaminated surfaces, often via the hands of healthcare workers[3], results in transmission to and colonization of a new vulnerable host[4]. Colonization is a strong predictor of future clinical infection[5–7]. The cycle repeats when the second patient becomes a new reservoir for onward nosocomial transmission.

✉e-mail: s.kanjilal@amsterdamumc.nl

Active surveillance for colonization of asymptomatic individuals is a key component of infection control programs, but requires significant investment in infrastructure and human resources. For this reason it is typically limited to intensive care units (ICUs) and other high-risk units, and to a few drug resistant or high-virulence organisms, such as methicillin resistant *Staphylococcus aureus* (MRSA) and vancomycin resistant *Enterococcus* species (VRE). Colonization pressure (CP), defined as the prevalence of an organism among patients in the ward into which a patient enters, has previously been shown to be associated with hospital acquired infection[8–10]. CP surveillance has the potential to augment active surveillance efforts as its estimation relies solely on information already present in the electronic health record (EHR). Additionally, it can be calculated for any number of organisms, in any area of the hospital, whereas expanding active surveillance can be disruptive to care and costly.

Prior analyses examining the impact of CP on hospital acquired infection were limited to known high-risk clonal pathogens and to ICU settings[10–15]. Whether the same relationship applies to drug susceptible organisms, which are at least as common as drug-resistant organisms[16,17], and to non-ICU settings, remains unknown. Thus, the objectives of this study are twofold. The first is to build a prototype of an open source infection control informatics tool that takes raw EHR data and constructs hospital unit-level CP across a variety of common organisms and to publish a de-identified patient level CP dataset for reproducibility and generalizability. Second, we use our prototype to test the hypothesis that CP is associated with hospital acquisition for a wide range of drug susceptible and drug resistant organisms relative to controls matched by demographics, length of stay, surgery, and fine-grained antibiotic exposures.

## Results

### Baseline characteristics were well matched between cases and controls

A total of 43,403 patients met inclusion criteria, with 14,923 cases of hospital acquired organisms matched to 28,480 controls. Sample sizes for individual target organisms ranged from 610 for vancomycin resistant *E. faecium* (244 cases matched to 366 controls), to 9,460 for methicillin susceptible *S. aureus* (MSSA, 2956 cases matched to 6504 controls). Nosocomial acquisition occurred nearly 4 times more frequently for drug susceptible organisms (*Escherichia coli*, *Klebsiella pneumoniae*, vancomycin susceptible *E. faecalis*, MSSA, drug susceptible *Pseudomonas aeruginosa*), than for drug resistant organisms (extended spectrum β-lactamase producing (ESBL) *E. coli*, ESBL *K. pneumoniae*, vancomycin resistant *E. faecium*, MRSA, drug resistant *P. aeruginosa*). Flow diagrams for individual cohorts are shown in Supplementary Table 2.

Baseline characteristics were similar across cases and controls for each target organism. Mean age for cases was 63.6 years (SE ± 0.2 years) and 63.9 years (SE ± 0.1 years) for controls. Ages ranged from 52.1 years (SE ± 0.5 years), and 54.7 years (SE ± 0.3 years) for MSSA cases and controls, respectively, to 70.7 years (SE ± 0.9 years), and 70.7 years (SE ± 0.8 years) for vancomycin resistant *E. faecium* cases and controls, respectively. Similarly, mean Elixhauser index ranged from 8.8 (SE ± 0.3) and 7.3 (SE ± 0.2) for MSSA case and controls, respectively, to 14.3 (SE ± 1.0), and 12.0 (SE ± 0.8) for vancomycin resistant *E. faecium* cases and controls, respectively. The number of prior antibiotic courses per 100 persons ranged from 3.5 and 2.8 for *E. coli* cases and controls, respectively to 9.0 and 7.7 for *Clostridioides difficile* cases and controls, respectively. Baseline features for the pooled cohort are shown in Table 1, and relevant characteristics for individual organism cohorts are shown in Fig. 1, and Supplementary Tables S3 to S7. The distribution of comorbidities by individual Elixhauser categories, class-level antibiotic exposures, and the cumulative distribution of colonization pressures are shown in Supplementary Fig. S1 to S5.

**Table 1 | Baseline characteristics in pooled cohort across all organisms**

| Feature | Pooled cohort | |
|---|---|---|
| | Cases | Controls |
| Sample size | 14,923 | 28,480 |
| Demographics | | |
| Age (mean, SE) | 63.6 +/− 0.2 | 63.9 +/− 0.1 |
| Gender (% female) | 55.9 | 56.1 |
| Elixhauser index (mean, SE) | 10.3 +/− 0.1 | 8.9 +/− 0.1 |
| Prior surgery (%) | 14.6 | 13.8 |
| Length of Stay (median, IQR) | 5.0 (3.2, 9.0) | 3.9 (2.9, 5.5) |
| Matching Duration (median, IQR) | 4.1 (3.1, 5.9) | 3.9 (2.9, 5.5) |
| Time to Infection (median, IQR) | 8.8 (5.0, 17.1) | -- |
| Prior antimicrobials (%) | | |
| β-lactams | | |
| Penicillins | 0.2 | 0.2 |
| Extended spectrum penicillins | 0.7 | 0.6 |
| Cephalosporins | 1.2 | 1.1 |
| Extended spectrum cephalosporins | 0.0 | 0.0 |
| Anti-Staphylococcal β-lactams | 0.0 | 0.0 |
| Other cell wall active agents | | |
| Glycopeptides | 0.0 | 0.0 |
| DNA synthesis inhibitors | | |
| Sulfonamides | 0.8 | 0.7 |
| Protein synthesis inhibitors | | |
| Fluoroquinolones | 1.1 | 1.0 |
| Tetracyclines | 0.6 | 0.5 |
| Macrolides | 0.4 | 0.4 |
| Lincosamides | 0.1 | 0.1 |
| Anti-anaerobic antibiotics | | |
| Metronidazole | 0.1 | 0.1 |
| Oral vancomycin | 0.0 | 0.0 |

### Colonization pressure is primarily directly associated with nosocomial acquisition among cognate pairs

The adjusted odds ratio (OR) of hospital acquisition of an organism due to cognate colonization pressures is shown in Fig. 2. Among the enteric flora, the strongest positive association was between $CP_{C.\ difficile}$ and hospital acquisition of *C. difficile* (OR 1.33, 95% CI 1.22 − 1.44). Similarly, $CP_{VRE}$ and $CP_{ESBL\ Enterobacterales}$ were both associated with acquiring ESBL *K. pneumoniae*, with ORs of 1.29 (95% CI 1.11 − 1.51), and 1.10 (95% CI 1.03 − 1.19), respectively. A unit increase in $CP_{vancomycin\ susceptible\ Enterococcus\ species\ (VSE)}$ was associated with an increase in the odds of acquiring five enteric organisms: ESBL *E. coli* (OR 1.05, 95% CI 1.01–1.09), drug susceptible *K. pneumoniae* (OR 1.03, 95% CI 1.00–1.06), *C. difficile* (OR 1.06, 95% CI 1.00–1.12), vancomycin susceptible *E. faecalis* (OR 1.12, 95% CI 1.09–1.15), and vancomycin resistant *E. faecium* (OR 1.09, 95% CI 1.01–1.17). Finally, a unit increase in $CP_{Enterobacterales}$ was associated with acquiring drug susceptible *E. coli* (OR 1.01, 95% CI 1.00–1.02).

$CP_{Enterobacterales}$ was associated with a protective effect for hospital acquisition from 5 enteric organisms, including ESBL *E. coli* (OR 0.98, 95% CI 0.96–0.99), ESBL *K. pneumoniae* (OR 0.96, 95% CI 0.93–1.00), *C. difficile* (OR 0.98, 95% CI 0.96–1.00), vancomycin susceptible *E. faecalis* (OR 0.98, 95% CI 0.97–0.99), and vancomycin resistant *E. faecium* (OR 0.96, 95% CI 0.93–0.99). Similarly, $CP_{C.\ difficile}$ had a protective effect for hospital acquisition of drug susceptible *E. coli* (OR 0.96, 95% CI 0.93–0.99), *K. pneumoniae* (OR 0.95, 95% CI 0.91–0.99) and vancomycin susceptible *E. faecalis* (OR 0.91, 95% CI 0.86–0.95).

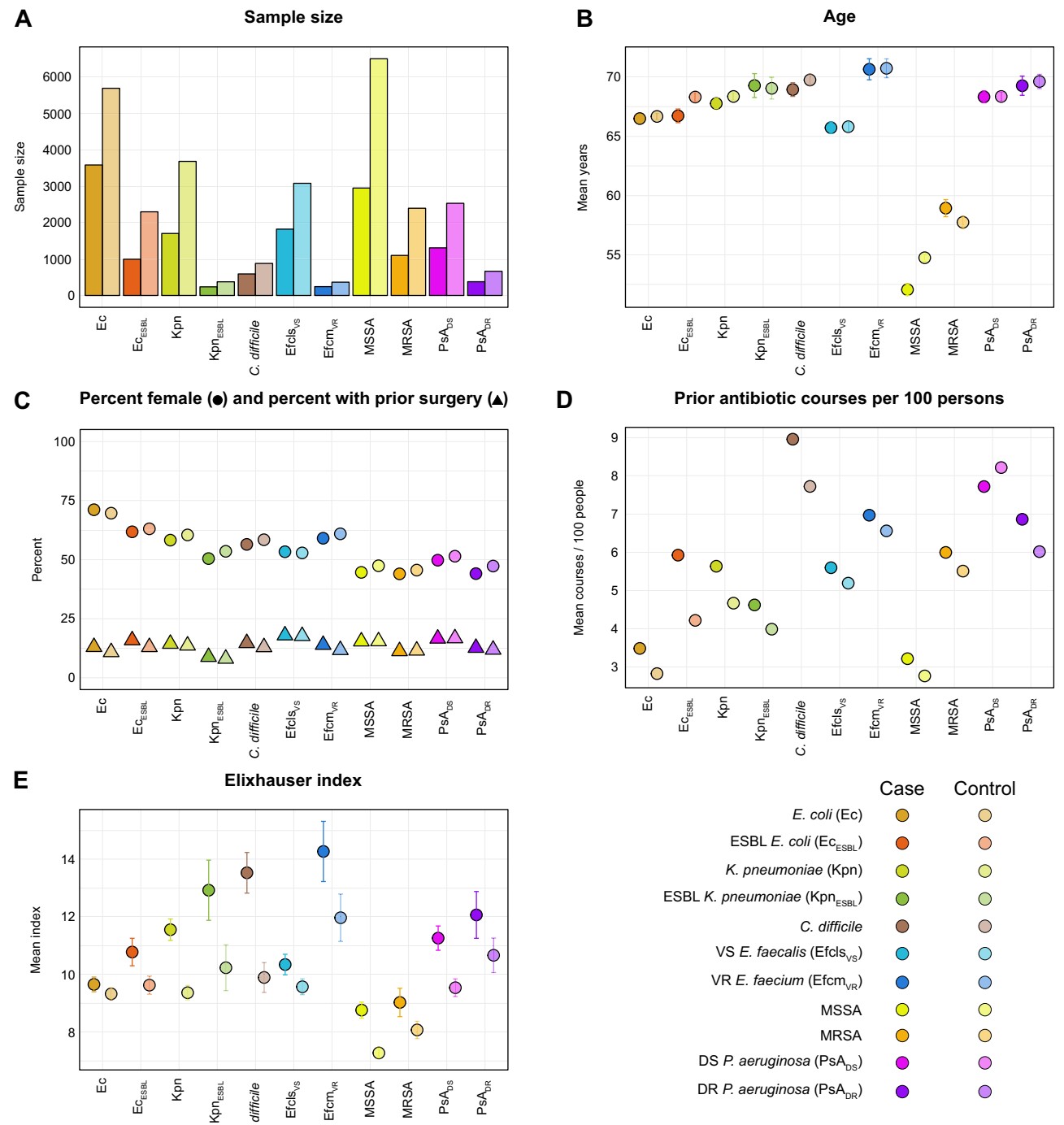

**Fig. 1 | Select baseline characteristics across 11 cohorts of target organisms.** Sample sizes are shown in panel **A** and provided in Supplementary Tables S3 to S7. Error bars represent standard errors of the mean. Cases and controls were matched by age (**B**), sex (**C**), history of any surgery in the previous 90 days (**C**), total number of antibiotic courses (**D**), and length of stay. Models included the Elixhauser Mortality Index (**E**). ESBL extended spectrum β-lactamase producing, VS vancomycin susceptible, VR vancomycin resistant, DS drug susceptible, DR drug resistant.

Among the skin flora, a unit increase in CP_MSSA was associated with nosocomial acquisition of MSSA (OR 1.12, 95% CI 1.10–1.14), and a unit increase in CP_MRSA was associated with nosocomial acquisition of MRSA (OR 1.07, 95% CI 1.01–1.13). Among environmental flora, CP_drug susceptible *P. aeruginosa* (PsA-DS) and CP_drug resistant *P. aeruginosa* (PsA-DR) were associated with nosocomial acquisition of drug susceptible *P. aeruginosa* (OR 1.10, 95% CI 1.05–1.15) and drug resistant *P. aeruginosa* (OR 1.29, 95% CI 1.14–1.45), respectively.

## Colonization pressure is primarily inversely associated with nosocomial acquisition among non-cognate pairs

Figure 3 shows the adjusted odds of hospital acquisition of an organism due to non-cognate colonization pressures. Eleven of 13 significant associations were protective in these models. For enteric organisms, a unit increase in CP_MRSA was negatively associated with drug susceptible *E. coli* (OR 0.95, 95% CI 0.92–0.98) and *K. pneumoniae* (OR 0.95, 95% CI 0.91–0.99), respectively. A unit increase in CP_MSSA was inversely

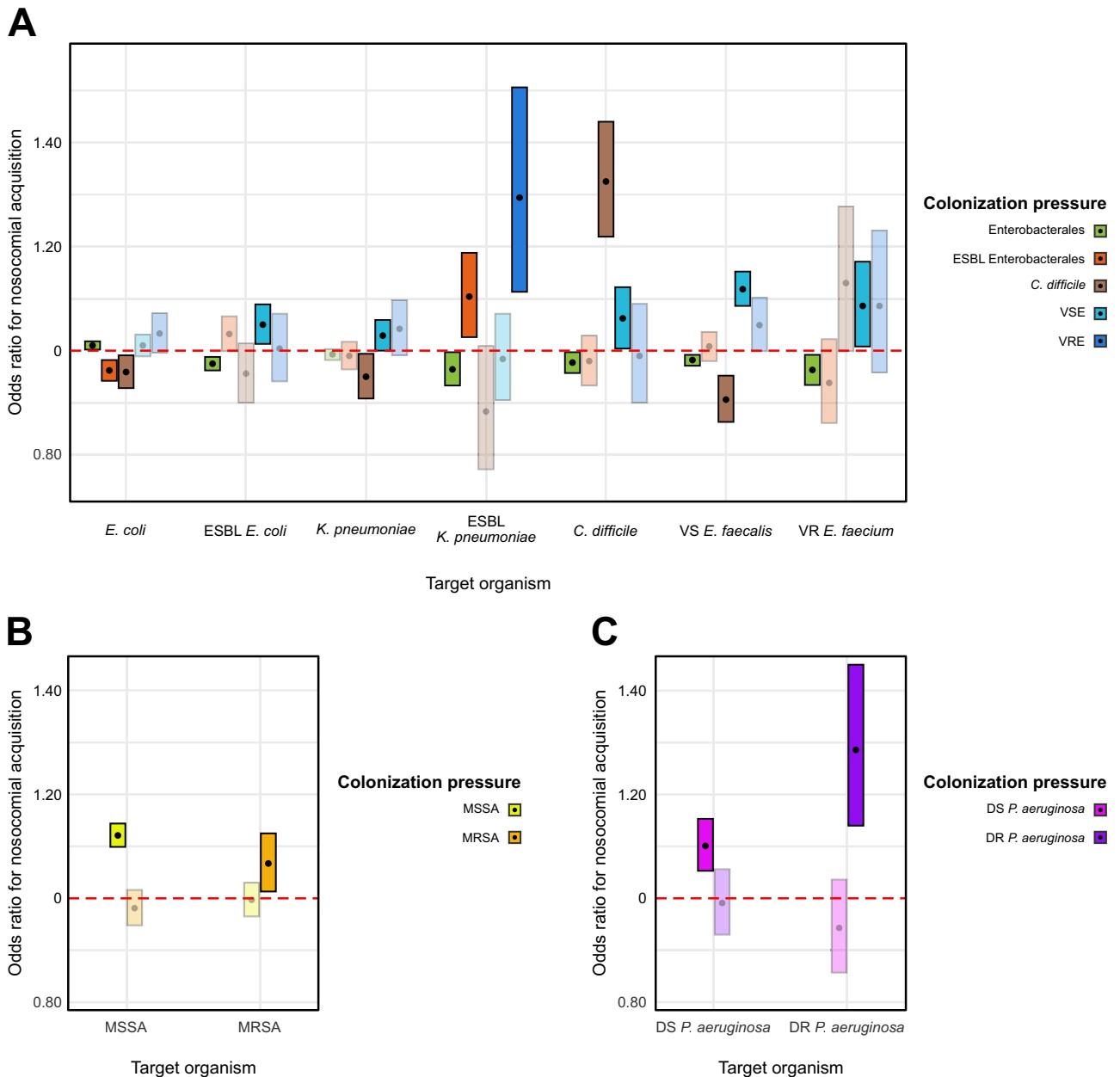

**Fig. 2 | Impact of colonization pressure on odds of nosocomial acquisition of target cognate organisms. A** Impact of CP from enteric organisms on hospital acquisition of target enteric organisms. **B** Impact of CP from skin flora on hospital acquisition of target skin flora. **C** Impact of CP from environmental flora on hospital acquisition of target environmental flora. Point estimates for effect size are given by black circles. Bars represent the 95% confidence interval around the point estimate. CP features with 95% CIs that do not cross 1.0 are considered statistically significant and are bolded. *Y*-axis scale identical across all panels. ESBL extended spectrum β-lactamase, VS vancomycin susceptible, VR vancomycin resistant, DS drug susceptible, DR drug resistant.

associated with ESBL *K. pneumoniae* (OR 0.92, 95% CI 0.85–1.00), while a unit increase in CP$_{PsA-DR}$ was inversely associated with vancomycin susceptible *E. faecalis* (OR 0.90, 95% CI 0.84–0.96).

Among the skin flora, the odds of hospital acquisition of MSSA were reduced for unit increases in CP$_{Enterobacterales}$ (OR 0.96, 95% CI 0.95–0.97), CP$_{ESBL\ Enterobacterales}$ (OR 0.96, 95% CI 0.94–0.99), and CP$_{C.\ difficile}$ (OR 0.94, 95% CI 0.90–0.98), while the odds of MRSA were reduced for unit increases in CP$_{Enterobacterales}$ (OR 0.98, 95% CI 0.96–0.99). For the environmental flora, the odds of drug susceptible *P. aeruginosa* decreased with unit increases in CP$_{Enterobacterales}$ (OR 0.99, 95% CI 0.97–1.00), CP$_{ESBL\ Enterobacterales}$ (OR 0.97, 95% CI 0.93–1.00), and CP$_{VRE}$ (OR 0.94, 95% CI 0.89–1.00). Two positive associations were noted, both related to unit increases in CP$_{VRE}$, MSSA

(OR 1.07, 95% CI 1.03–1.11) and drug resistant *P. aeruginosa* (OR 1.11, 95% CI 1.00–1.23). Supplementary Table S8 contains the coefficients for all conditional logistic regression models.

## Interpretable effects of colonization pressure

To better understand the impact of colonization pressure in concrete terms, we estimated the number of ward co-occupants associated with a unit change in CP (Fig. 4). Assuming median times since infection of 30, 90 and 180 days, a 1 unit increase in CP requires 1.4, 2.3 and 3.6 ward co-occupants to have had the CP organism, respectively. We then estimated the number of ward co-occupants with prior acquisition from each of the 9 CP organism sets, for each of the 11 target pathogen cohorts, differentiating between cases and controls, and stratifying by

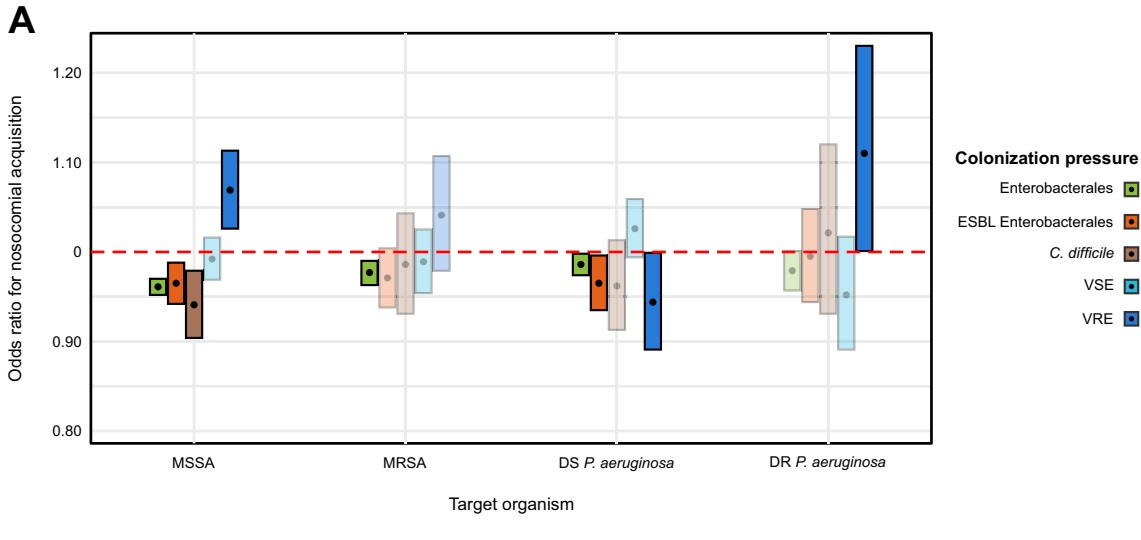

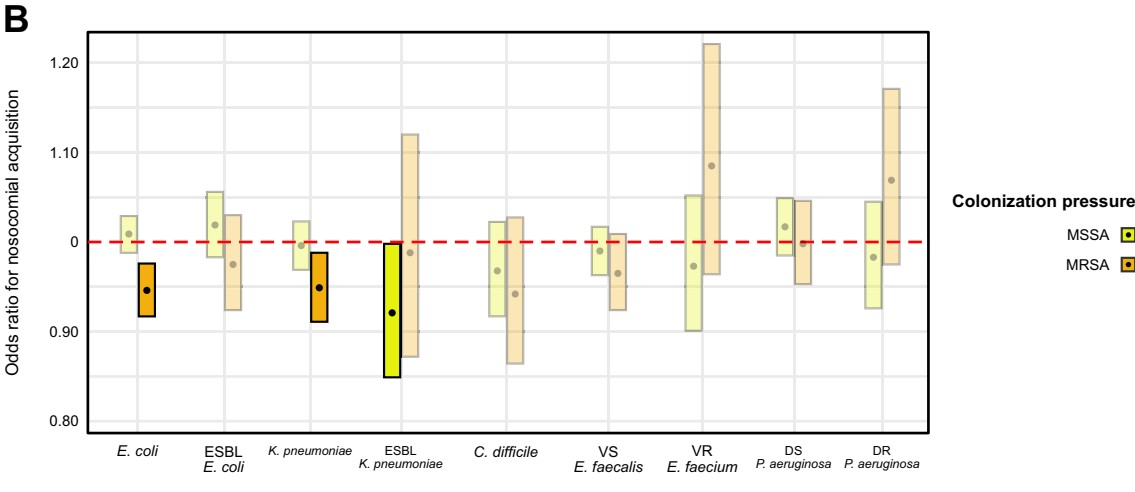

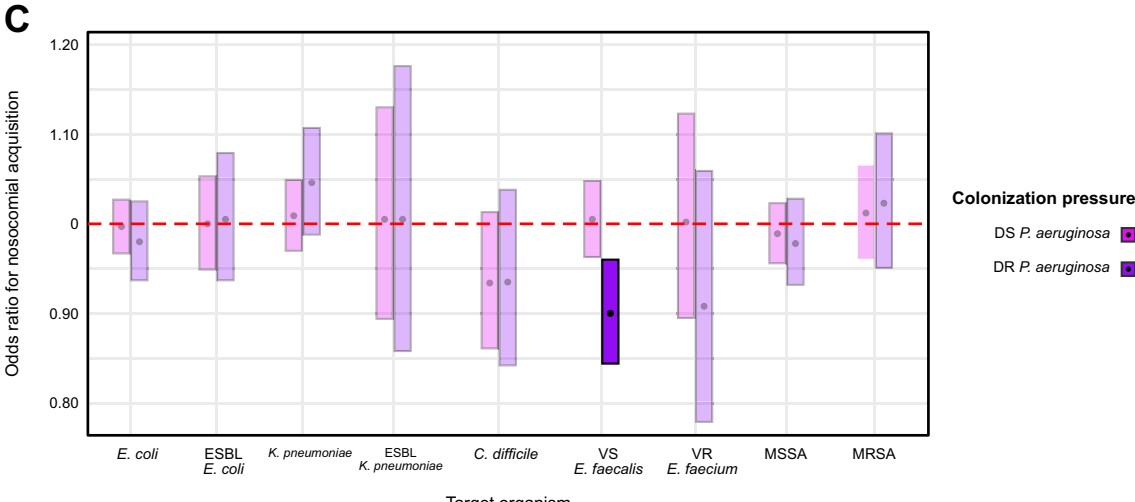

**Fig. 3 | Impact of colonization pressure on odds of nosocomial acquisition of target non-cognate organisms. A** Impact of CP from enteric organisms on hospital acquisition of target skin and environmental organisms. **B** Impact of CP from skin flora on hospital acquisition of target enteric and environmental organisms. **C** Impact of CP from environmental flora on hospital acquisition of target enteric and skin organisms. Point estimates for effect size are given by black circles. Bars represent the 95% confidence interval around the point estimate. CP features with 95% CIs that do not cross 1.0 are considered statistically significant and are bolded. *Y*-axis scale identical across all plots. ESBL extended spectrum β-lactamase producing, VS vancomycin susceptible, VR vancomycin resistant, DS drug susceptible, DR drug resistant.

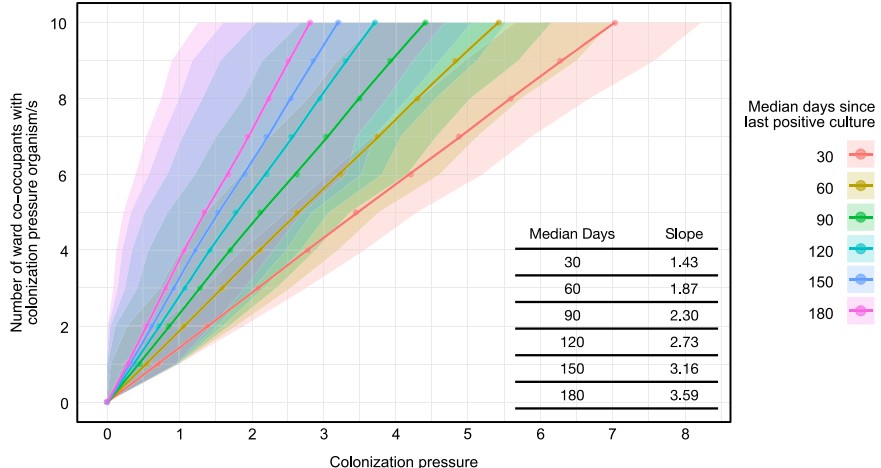

**Fig. 4 | Number of ward co-occupants with a colonization pressure organism or organisms by colonization pressure values.** Estimates are agnostic to organism and are stratified by the median number of days since ward co-occupants' last culture was positive for the colonization pressure organism/s. Variance estimates drawn from 100 bootstrapped simulations.

median time to infection (supplemental fig. S6 through S10). As an example, the greatest positive effect size was for $CP_{C.\ difficile}$ and hospital acquisition of *C. difficile* (OR 1.33, 95% CI 1.22–1.44). Given the median $CP_{C.\ difficile}$ of 1.46 for the cohort of *C. difficile* cases, 2.1, 3.4 and 5.3 ward co-occupants would be estimated to have had acquired *C. difficile*, a median of 30, 90 and 180 days prior to the entry of cases into the ward, respectively, compared to 1.2, 2.0 and 3.2 ward co-occupants in the control group. Similarly, the greatest negative effect size was for $CP_{PsA-DR}$ and vancomycin susceptible *E. faecalis* (OR 0.90, 95% CI 0.84–0.96). In the cohort of cases with vancomycin susceptible *E. faecalis*, the median $CP_{PsA-DR}$ was 0.69, which translates to 1.0, 1.6 and 2.6 ward co-occupants with acquisition of drug resistant *P. aeruginosa*, a median of 30, 90 and 180 days prior to the entry of cases into the ward, respectively, compared to 1.1, 1.7 and 2.8 ward co-occupants in the control group. These values provide a point of reference for the impact of CP on target pathogens in terms of numbers of patients.

### XGBoost models using colonization pressure do not predict nosocomial acquisition

AUROCs for XGB models using CP and Elixhauser index to predict nosocomial acquisition ranged from 0.56 (95% CI 0.49 to 0.62) for ESBL *K. pneumoniae* to 0.63 (95% CI 0.60 to 0.67) for *C. difficile*. Positive predictive values ranged from 33% for MRSA to 50% for *C. difficile*, and negative predictive values ranged from 61% for drug resistant *P. aeruginosa* to 75% for MSSA. Likelihood ratio positive values ranged from 1.05 for MRSA to 1.54 for *C. difficile*, while likelihood ratio negative values ranged from 0.97 for MRSA to 0.68 for *C. difficile*. The full distribution of metrics for the conditional logistic regression and XGBoost prediction models is found in Supplementary Table S9. Mean SHAP values for conditional logistic regression and XGB models are located in Supplementary Table S10 and SHAP plots are located in Supplementary Fig. S11 through S16.

## Discussion

Using routinely collected EHR data at a large tertiary care hospital system with advanced infection control practices, we show that a patient's odds of acquiring a potential pathogen during a hospitalization is associated with the burden of infection among that patient's ward co-occupants. Associations were noted for drug resistant as well as drug susceptible organisms, between those that typically inhabit the same as well as distinct niches, and included direct as well as indirect relationships. Our proof-of-concept study reinforces the hypothesis

that colonization pressure is a proxy of a unit's microbiome and opens up new avenues of research for microbial ecology and hospital epidemiology with the potential to augment existing infection control practices.

There is a large body of work describing the risks of infection conferred by the built environment and healthcare delivery. Metagenomic investigations into the biogeography of indoor environments and hospital rooms point to bidirectional transmission between room occupants and their room[18–21], with adverse clinical outcomes of room-to-occupant transmission concentrated in certain vulnerable sub-populations such as neonates[22,23]. Our expanded scope of analysis suggests room-to-occupant transmission may be widespread throughout the hospital. Health care epidemiology analyses provide a complementary perspective by defining the role of healthcare workers as vectors for transmission[3,21,24] and quantifying individual level risk factors for hospital acquired infection[25]. While both fields have yielded critical insights, a common weakness is the use of variables that cannot be easily operationalized for real time surveillance.

While it is difficult to directly compare our study to others due to differences in methods, our results are largely in line with prior research examining colonization pressure. Lawrence et al. defined *C. difficile* ICU CP as the sum of a daily point prevalence, and found a statistically significant rate of acquisition when CP crossed a threshold of 10 patient-days of exposure[26], which is comparable to our estimate of 2 to 5 ward co-occupants with prior infection between 30 and 180 days in the past (Supplementary Fig. S8). Similarly, Jolivet et al. defined CP as the ratio of colonized ICU patient-days to total ICU patient-days and found a threshold of 100 to 200 patient days to be associated with odds of acquisition for ESBL Enterobacterales[12], which is an order of magnitude weaker than for *C. difficile*. This is within range of our estimates that show ESBL *K. pneumoniae* acquisition to be associated with a CP of between 4 and 12 ward co-occupants having ESBL Enterobacterales 30 to 180 days prior (Supplementary Fig. S7). Finally, Chanderraj et al. used a matched case control design to examine the influence of VRE CP on VRE acquisition in the ICU, in addition to patient-level factors such as antibiotics and proton pump inhibitor exposure, and found no association[15]. This is in agreement with our findings that VRE acquisition was not associated with CP from VRE, though it was associated with CP from VSE. Importantly, all of these studies were performed in ICUs, which are mostly single room occupancy, and none performed matching to equalize prior antibiotic exposures. To our knowledge, no studies have examined relationships

between CP and drug susceptible organisms or non-cognate pairs, and no studies have found inverse associations in multivariate models.

High-risk clones are thought to be successful due to the accumulation of phenotypic traits that confer a survival advantage under the selective pressure of the hospital environment. These elements include resistance to antimicrobials and disinfectants, the ability to form biofilms and the expression of virulence factors that permit survival under pressure from microbial competitors and the host immune system[27]. Our findings suggest that while these factors may lead clones to transmit more efficiently than strains that lack them, they may not be necessary for hospital acquisition, particularly when the burden of colonization within a unit is high. It is notable that despite having weaker associations, the absolute burden attributable to hospital acquired drug susceptible organisms was several times greater than that due to drug resistant organisms in our study. This may be the result of a lack of targeted infection control measures for this organism group. In our study, this is reflected by the higher mean colonization pressures for drug susceptible organisms (Enterobacterales, VSE, MSSA, drug susceptible *P. aeruginosa*) than for drug resistant organisms (ESBL Enterobacterales, VRE, MRSA and drug resistant *P. aeruginosa*).

Nosocomial acquisition for several organisms had an inverse relationship with colonization pressure, meaning model-adjusted CP values were higher in controls than in cases. This was apparent for both cognate and non-cognate pairs, though it occurred more often in the latter. There are several potential explanations for this. The first two involve confounding between comparators due to differences in ward characteristics or patient characteristics. For instance, if cases were on wards with fewer double occupancy rooms[28], or higher rates of contact precautions, then rates of nosocomial acquisition would be expected to be lower, depending on the baseline rate of colonization in the unit. A similar outcome could be seen if a higher proportion of controls had unmeasured patient-specific features associated with nosocomial acquisition, such as a specific type of surgery, hardware or intubation. Though only true randomization would eliminate bias entirely, both of these scenarios are somewhat unlikely as we applied stringent matching criteria, including across 14 classes of antibiotics. This serves two purposes. First, it equalizes selective pressures that would promote differences in silent carriage, and second, because single and combination antibiotic therapies have a finite number of indications, fine-grained matching effectively clusters similar types of patients together onto similar wards. This is supported by the similar distributions of individual Elixhauser categories between cases and controls, despite not being included in the match.

A third possible explanation for the inverse relationship is that contamination of a ward by one organism may prevent other organisms from establishing themselves. This may be due to repeated contamination by colonized hosts on a given floor, or alternatively, intra- and inter-species competition on hospital environmental surfaces. The concept of colonization resistance and competition has been described in vitro and for niches across the human body[29–32], but further research is necessary to prove whether founder effects and other competitive mechanisms take place on high touch surfaces, sinks, and bathrooms within hospital units.

In addition to supporting reproducibility, our goal with releasing an extensively annotated analytic pipeline and de-identified patient-level data is to allow other hospital systems to adapt and improve on our methods. Our hope is for the research community to collectively build a generalizable, automated, machine learning-based, real-time risk assessment tool for nosocomial pathogen acquisition that functions across the hospital, for any range of organisms, and eventually across different health systems. Such a tool could be used to study colonization pressure on a broader scale, which we hope will generate an evidence base for pre-emptive measures to prevent infections or enact closer surveillance on wards that cross a pre-designated threshold. It is important to note however, that despite the

advantages of using a machine learning approach to maximize predictive power, our XGBoost models had low positive and negative predictive value for hospital acquisition. This may be explained by small sample sizes and the fact that colonization pressure is one of many factors that must be included in real-time models to support decision-making. In its current form, our tool and models represent a proof-of-concept and require further refinement before they are ready to augment existing infection control interventions.

The other major strength of this study is our incorporation of a wide range of common organisms into our CP models. This allows us to estimate the odds of hospital acquisition for a given organism while accounting for the complex ecology of a hospital unit, and to identify interactions that may be occurring across species and niches. To our knowledge, this is the first study to utilize this strategy. Future work aims to improve our model's predictive performance by addition of host-specific factors and integration with clinical pathogen whole genome sequencing data to better define transmission events. Model performance will be improved by including data from other hospital systems, which will bolster sample size and generalizability.

Our study has a number of limitations. Our modest sample sizes restrict the number of features that could be included in the models. This reflects a decision to apply very strict inclusion criteria for cases and controls, thereby providing the cleanest possible comparison for estimating the impact of hospital environment on nosocomial acquisition using observational data. For instance, the use of highly granular matching of antibiotic exposures by both class and burden of exposure helps ensure that selective pressures from systemic antimicrobials are closely aligned for cases and controls. Restricting the analysis to patients with a single room stay prevents confounding that may occur if the patient was exposed to another hospital unit during the same hospitalization, but leads to reduced generalizability as patients often transfer between wards, such as those admitted to the ICU. As noted earlier, our results represent a proof-of-principle that CP impacts nosocomial acquisition, but we cannot extrapolate its importance relative to other factors with this study.

Another limitation is our inability to identify silent carriage of a target organism, which could confound the relationship between CP and hospital acquisition. This stems partly from the fact that many of our target organisms do not have surveillance programs for asymptomatic sampling, thus our insight into the true burden of disease is dependent on recognition by providers, which is highly variable. Additionally, we lack insight into events (microbiology specimens, antibiotic prescriptions) that occur outside of our healthcare system. We have reduced this possibility by excluding patients with evidence of the target organism in the previous 6 or 12 months and our antibiotic exposure matching strategy. Finally, our data come from a single healthcare system in the northeast US and caution is warranted when extrapolating to other geographic areas and healthcare systems that may differ in terms of patient demographics, clinical sampling practices, antibiotic consumption, and infection control policies. In particular, as our hospital contains a mix of single and multi-occupant rooms, these results may not generalize to newer hospitals that contain solely single occupant rooms. This could be surmounted by including occupancy data and combining data across multiple health systems using a common data model or a federated learning approach.

In conclusion, our data affirm the importance of colonization pressure at the hospital unit level as a risk factor for acquisition of common pathogens. We quantify the effect for 11 drug susceptible and drug resistant organisms, inhabiting 3 distinct niches, and identified direct and inverse relationships with 9 CP organism sets. While more easily treated with antibiotics, the absolute burden of hospital acquired drug susceptible organisms in our health system was four times higher than that for the organisms typically surveilled by infection control programs. Further research is necessary to improve prediction models using CP, and understand the costs and benefits of

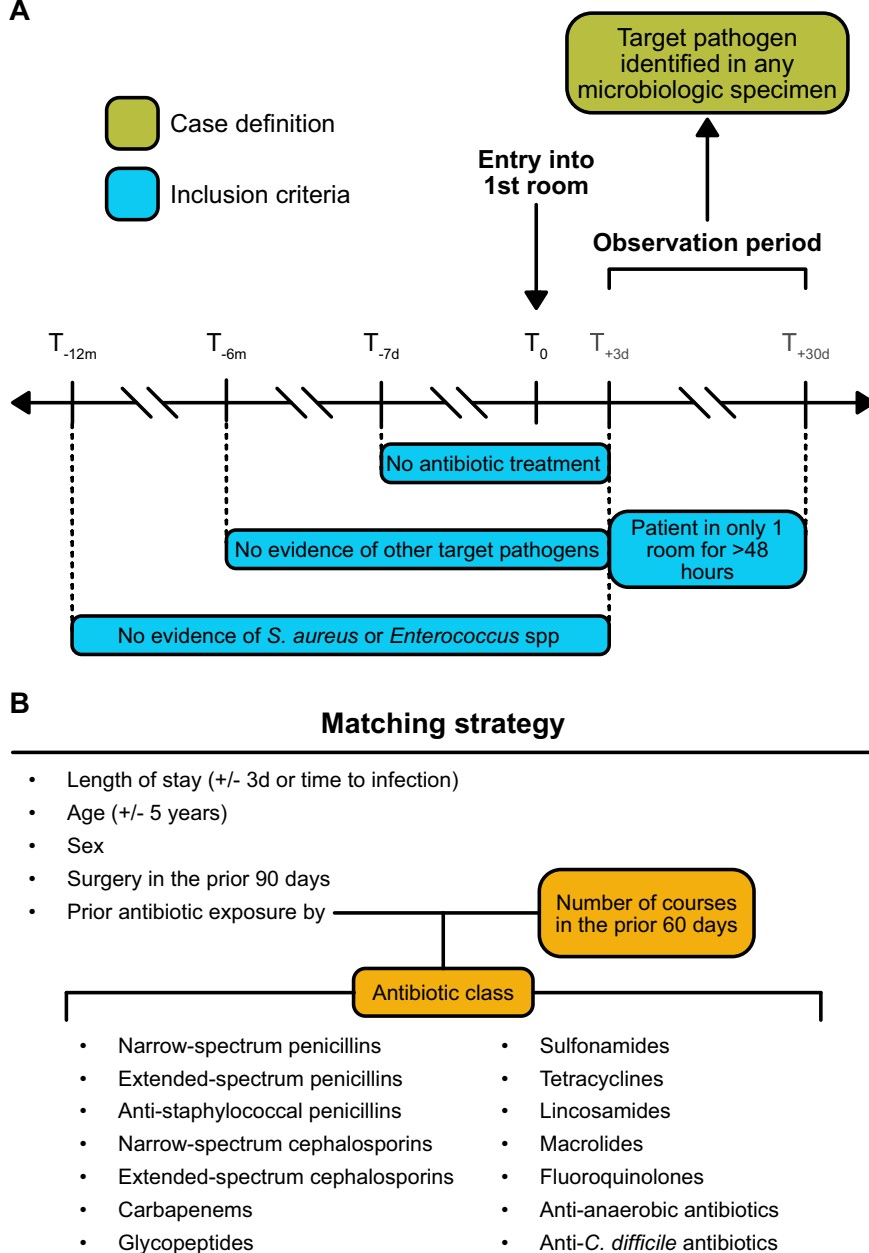

**Fig. 5 | Study design. A** $T_0$ was set as the time stamp of entry into the room for cases and controls. This room always represented the first room in a patient's hospital stay that lasted >48 h, not including the emergency room. All study participants were observed for culture positivity with the target organism between day +3 to day +30 after entry into their first room. All study participants had no evidence of antibiotic exposure in the previous 7 days, nor any clinical or surveillance cultures with a target organism in the previous 6 months (12 months if the target organism was *S. aureus* or an *Enterococcus* species). They also stayed in no more than one room for >48 h during the observation period. **B** Matching strategy of cases to controls. Cases were matched to controls based on age, sex, history of surgery in the previous 90 days, length of stay, and antibiotic exposure. The latter was further stratified by the number of courses consisting of ≥2 days of exposure that were >7 days apart, in the previous 60 days, across 14 drug classes.

applying real-time surveillance of CP to augment existing hospital infection control practices.

## Methods

This study was approved by the Institutional Review Board of Mass General Brigham. We have adhered to STROBE and RECORD reporting guidelines.

### Study design

We conducted a series of matched case-control studies using data obtained from the Mass General Brigham (MGB) healthcare system's EHR data warehouse. MGB includes 8 community and 2 tertiary care hospitals in the New England area. Patients were included in the study if they were adults (>=18 years old) admitted to any MGB hospital between May 25 2015 and July 7 2024.

### Cohort definition

A case was defined as a patient with a clinical or surveillance specimen obtained from any body site with the target organism during the observation period, which was set as day +3 to day +30 after entry into their inpatient room. Controls were matched 2:1 by age ±5 years, sex, and history of any surgery in the previous 90 days. Controls were also required to have a length of stay in their room that was within ±3 days of a case's length of stay, or the date timestamp of collection for the

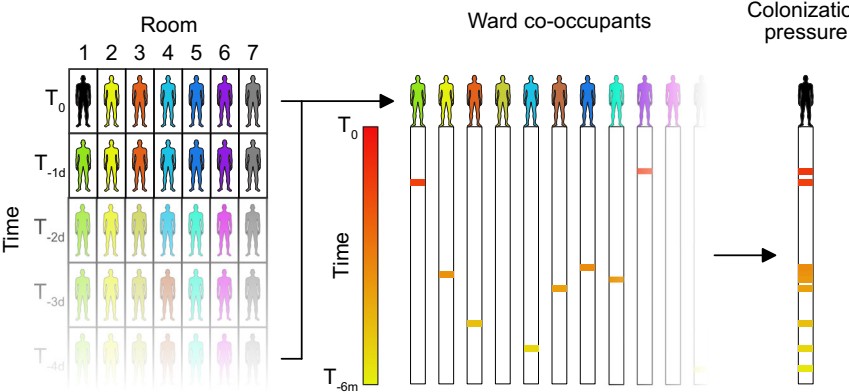

**Fig. 6 | Definition of colonization pressure.** Colonization pressure (CP) was defined as the time-weighted prevalence of a specified organism, or organism sets, in ward co-occupants. In the figure, the case or control enters Room 1 at $T_0$. CP was calculated by identifying ward co-occupants in the previous 30 days and recording the number of days since they had the CP organism (or organism set) in the previous 6 months. A negative log transform was then applied to the number of days to account for diminishing effect over time and values were summed across all ward co-occupants to calculate the final value.

case's index culture, whichever was shorter. For example, if a case was in a room for 20 days and the culture with the target pathogen was identified on day 12, the matched control was in their room between 9 to 15 days. Finally, controls were matched on the number of courses of antibiotics (inpatient and outpatient) received in the previous 60 days. A course was defined as >2 days of treatment >7 days apart and was stratified by the following antibiotic classes: penicillins, anti-staphylococcal penicillins, extended-spectrum penicillins, cephalosporins, extended-spectrum cephalosporins, carbapenems, glycopeptides, fluoroquinolones, macrolides, tetracyclines, anti-anaerobic antibiotics and a separate category for oral vancomycin for targeting *C. difficile*. A full list of antibiotics within each treatment class category is located in Supplementary Table S1.

Both cases and controls were required to have stayed in only one room for >48 h during the entirety of the observation period. Collection of the index culture after departure from the room was allowed if the patient had no other stay for >48 h between day +3 to day +30. Other study inclusion criteria included not being actively treated for infection, defined as no record of antibiotic exposure from day −7 to day +3 relative to the time of entry into their room. Both cases and controls also had to have no prior evidence of the organism of interest in the previous 12 months for *S. aureus* and *Enterococcus* species, and the previous 6 months for all other target organisms. Figure 5 depicts the study design.

### Target organisms
We built cohorts to predict nosocomial acquisition for 11 different organisms across skin, enteric and environmental niches. Among enteric organisms, we selected drug susceptible and ESBL *E. coli* and *K. pneumoniae*, vancomycin susceptible *E. faecalis*, vancomycin resistant *E. faecium*, and *C. difficile*. A drug susceptible *E. coli* and *K. pneumoniae* phenotype was defined as an organism susceptible to 3rd and 4th generation cephalosporins and piperacillin-tazobactam, while the ESBL-producing phenotype was defined as resistance to any 3rd or 4th generation cephalosporin or piperacillin-tazobactam. Both drug susceptible and ESBL Enterobacterales phenotypes retained susceptibility to carbapenems. Among skin organisms, we selected MSSA and MRSA. Among environmental organisms, we selected drug susceptible *P. aeruginosa*, defined as an organism susceptible to ceftazidime, cefepime and piperacillin-tazobactam, and drug resistant *P. aeruginosa*, defined as resistance to any one of the anti-Pseudomonal β-lactams.

### Model covariates
We utilized colonization pressure (CP) to estimate the impact of the hospital environment on the odds of a hospital acquisition. We defined CP as the time-weighted prevalence of a specified organism, or organism set (as in the Enterobacterales), in ward co-occupants, given by Eq. (1):

$$Study\,participant_{ward_i} = \sum_{Co-occupant_1}^{Co-occupant_j} Y_j \cdot e^{(-\lambda T_j)} \tag{1}$$

where $Y_j = 1$ if the ward co-occupant had the target organism within the past 6 months ($Y_j = 0$ if not), $T_j$ is the time interval in days between the date the organism was identified in the co-occupant and the date of entry into the ward by the study participant (case or control), and λ is a tunable decay parameter controlling the influence of time (set to 0.01 by default). A ward co-occupant was defined as a person with a > 48 h stay in the 30-day period prior to entry into the ward by the study participant. The look-back period for $Y_j$ and $T_j$ was set to 6 months to account for the possibility of organism carriage. A negative log transform was applied to $T_j$ to account for the diminishing effect of prior organisms over time. CP was calculated separately for the following 9 organisms and groups: drug susceptible Enterobacterales and ESBL Enterobacterales, vancomycin susceptible (VSE) and vancomycin resistant *Enterococcus* (VRE) species, *C. difficile*, MSSA, MRSA, and drug susceptible (DS-PsA) and drug resistant *P. aeruginosa* (DR-PsA). Drug susceptible, drug resistant, and ESBL phenotypes are defined as above for the target organisms. Figure 6 illustrates the calculation of CP. Models also included patient comorbidities, encoded by the Agency for Healthcare Quality Elixhauser Mortality Index[33].

### Statistical analyses
We utilized conditional logistic regression for estimating the impact of organism-specific CP on the odds of nosocomial acquisition. 95% confidence intervals were determined using the Wald method, as implemented in the survival R package. Statistical significance was determined based on whether the confidence interval around exponentiated coefficients crossed 1. The performance of the conditional logistic regression models was evaluated using the area under the receiver operator characteristic (AUROC) separately for each target organism dataset. 95% confidence intervals for AUROCs were calculated using the standard deviation across five bootstrapped validation folds. We report results separately for cognate relationships, defined as the target organism and the CP organism/s belonging to the same niche (e.g., skin, enteric, or environmental), and non-cognate relationships, where they inhabit different niches.

XGBoost models[34] using the same covariates were used to further verify the result of the conditional logistic regression. The performance of XGBoost models was also evaluated through 5-fold cross-validation. In each fold, the full dataset was divided into training, validation, and test sets, with a ratio of 8:1:1. The performance was evaluated on the held-out test set in each fold, and the 95% confidence interval of the AUROC was estimated using the standard deviation of the AUROC across the five folds.

A grid search was used to determine the optimal hyperparameters of the conditional logistic regression and XGBoost models. Feature importance was determined using the exponentiated coefficient of individual features for the conditional logistic regression and with Shapley values[35] for XGB models. All analyses were performed using R (version 4.4.0) and Python (version 3.6.15).

### Reporting summary

Further information on research design is available in the Nature Portfolio Reporting Summary linked to this article.

## Data availability

To promote transparency and reproducibility, we have made a de-identified version of the final cleaned patient-level data available under restricted access for the protection of human subjects and to ensure compliance with relevant data privacy regulations at Physionet[36,37]. Access can be obtained by registering for a PhysioNet account, completing the required CITI Program training in human subjects research (Data or Specimens Only Research), and signing a Data Use Agreement (DUA). The raw patient-level data are protected and are not available due to data privacy laws.

## Code availability

The source code for the entire analytic pipeline, spanning pre-processing of raw data to model output, is available on Github[38].

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

## Acknowledgements

This study was supported by the NLM grant 2T15LM007092-31 (LWS), AHRQ grant K08-HS030118 (TRP), AHRQ grant K08 HS027841-01A1 (S.K.), and institutional funding from the Department of Population Medicine (Z.W.). The authors would like to thank the Mass General Brigham Data Security and Privacy Office and the Institutional Review Board for permitting the posting of the de-identified dataset associated with this study.

## Author contributions

S.K. conceptualized the study. L.S., Z.W., C.M., C.C., A.A. and S.K. had access to the raw data and verified its quality. L.S., Z.W., C.M., C.C., A.A., T.P. and S.K. assisted with data preprocessing and curation. L.S. and Z.W. performed statistical modeling, and S.K. designed all data visualizations. S.K. wrote the original draft of the paper. L.S., Z.W., S.K., T.P., C.R., and M.K. assisted with reviewing and editing manuscript drafts. L.S., Z.W. and S.K. wrote the open source code base and Z.W. de-identified the data for online publication. SK supervised all aspects of the project.

## Competing interests

The authors declare no competing interests.

## Additional information

Luke Sagers[1,7], Ziming Wei[1,2,7], Caroline McKenna[2], Christina Chan[2], Anna A. Agan[2], Theodore R. Pak ®[2,3,4,5], Chanu Rhee[2,5], Michael Klompas ®[2,5] & Sanjat Kanjilal ®[2,5,6] ✉

[1]Department of Biomedical Informatics, Harvard Medical School, Boston, MA, USA. [2]Department of Population Medicine, Harvard Medical School and Harvard Pilgrim Health Care Institute, Boston, MA, USA. [3]Division of Infectious Diseases, University of California, Irvine School of Medicine, Irvine, CA, USA. [4]Division of Infectious Diseases, Massachusetts General Hospital, Boston, MA, USA. [5]Division of Infectious Diseases, Brigham & Women's Hospital, Boston, MA, USA. [6]Department of Medical Microbiology and Infection Prevention, Amsterdam UMC, University of Amsterdam, Amsterdam, the Netherlands. [7]These authors contributed equally: Luke Sagers, Ziming Wei. ✉e-mail: s.kanjilal@amsterdamumc.nl

