## [Transparent Peer Review file · Nature Communications]

Using electronic health records to assess the relationship between colonization pressure and nosocomial pathogen acquisition

Corresponding Author: Dr Sanjat Kanjilal

Version 0:

Reviewer comments:

Reviewer #1

(Remarks to the Author)

This is an interesting paper evaluating the colonization pressure for multiple pathogens in a healthcare system. Unfortunately, I am not able to access the supplementary materials so I cannot fully review this manuscript.

This sentence is confusing to me, "Controls were also required to have a length of stay in their room that was within ± 3 days of a case's length of stay, or the date timestamp of collection for the case's index culture, whichever was shorter." Could an example be given?

Is there a differential bias of this inclusion criterion, "stayed in only one room for >48 hours during the entirety of the observation period"? How did this impact the data included in the analysis?

On page 5, is it "never had the organism" or "never had the organism identified"?

Could the authors comment on the impact of their definition of a course of antibiotics, "A course was defined as >2 days of treatment >7 days apart and was stratified by the following antibiotic classes"? What proportion of patients received 2 or fewer days of antibiotics? Is the misclassification differential?

Similarly, what was the impact of this inclusion criterion, "Other study inclusion criteria included not being actively treated for infection, defined as no record of antibiotic exposure from day -7 to day +3 relative to the time of entry into their room."

The authors state that they adhered to STORBE, however, they provide limited information on participants, in particular items 13a+b (copied below). As I cannot see the supplemental files I'm not sure what is included in supplemental table 2 but there are insufficient data in the main text to evaluate this.

- (a) Report numbers of individuals at each stage of study—eg numbers potentially eligible, examined for eligibility, confirmed eligible, included in the study, completing follow-up, and analysed
- (b) Give reasons for non-participation at each stage
- (c) Consider use of a flow diagram

(Remarks on code availability)

I am not able to access the supplemental files and therefore cannot fully vet the code.

Is there cross validation happening in the conditional logistic regression? For the conditional logistic regression it doesn't appear that the data is subset at all. Are AUROCs reported for the LR in the supplement (which I cannot access)?

Is the data being split 80/10/10 for the xgboost modeling? For xgboost the data only appears to be split 80/20 with the test

set being used for both testing and validation. But this isn't where they are choosing the model hyperparameters so this code might not be shared/a version issue. Adding to that assumption, the output files from this script are being saved to another folder than the data is loaded from for plotting the final results.

Reviewer #2

(Remarks to the Author)

Sagers et al have developed an open-source informatics tool to estimate hospital unit-level colonisation pressure from EHR data, and used it to model nosocomial acquisition risk across a broad range of both drug-susceptible and drug-resistant pathogens. I believe this study is notable for its methodological thoroughness, its inclusion of a wide spectrum of organisms, large number of included patients, and its transparent, reproducible design supported by public release of data and code. I have also reviewed the code.

I would have liked continuous line numbers to be better able to refer to parts of the manuscripts, but this might be a limitation of the Nature submission platform? In any case, these are my comments of which I believe would increase the quality and clearness of the paper once addressed.

My background is in both clinical microbiology and data science. Therefore, I understand quite well how important efforts like the proposed approach are and I'm always happy to read about them in new submissions. Although:

- I feel the authors could have been considerably more explicit about the limitations imposed by relying solely on routine diagnostic data. While I understand and appreciate the rationale for using EHR-based surveillance, it would strengthen the discussion to more directly acknowledge the selective nature of clinical testing and its variation across hospitals and regions. In my view, it is important to recognise that while such modelling may complement existing surveillance infrastructure, it cannot substitute for active surveillance in terms of sensitivity or comprehensiveness.
- Given that all data were derived from a single integrated healthcare system in the northeastern United States, I feel the authors could be more forthright about the limits of generalisability. While the release of code and data is commendable and promotes reproducibility, institutional variation in diagnostic practices, antimicrobial stewardship, and infection control protocols could significantly impact both colonisation pressure and acquisition risk. This ought to be acknowledged more prominently, since it cannot be known how effective this model/approach would be in other sites. In other words, the study lacks external validation. Given the journal quality, I would propose that the authors either (1) add a complete external validation using another hospital site, which I understand is probably unfeasible, or (2) add to the manuscript that the paper describes a proof-of-concept rather than a validated approach. This should at least be clear in the title and/or abstract.
- Reading the section on XGBoost performance (which, I must admit, was somewhat disappointing - didn't expect a great decision tree model such as XGBoost to perform that bad), I found the relatively poor discriminatory capacity somewhat downplayed. While I agree that CP may be only one of many necessary features for accurate prediction, the limitations of the ML approach in its current form should be more clearly acknowledged. In particular, I would suggest the authors guard against overinterpreting the current utility of these models for real-time risk stratification.
- The paper makes a strong case for expanding the concept of colonisation pressure to include drug-susceptible organisms and non-ICU wards. Still, I would have appreciated a more direct comparison with prior work, especially in terms of quantitative effect sizes. In my view, situating this study more firmly within the context of prior colonisation pressure studies would clarify what is genuinely novel versus what confirms established findings in a broader setting.
- I was pleased to see that the authors have decided to make both their codebase and de-identified dataset publicly available. In my view, this represents a strong commitment to transparency and reproducibility, and sets an example that others in the field would do well to follow. However, at PhysioNet (<https://physionet.org>), I could not find the project entitled 'Predictors of hospital onset infection: A matched retrospective cohort dataset' as the authors indicate, or when searching for parts of that or for the manuscript title. Importantly, PhysioNet provides DOIs - please explicitly provide it in the manuscript to prevent others for searching in vain.

Lastly, I feel the title does not encompass the strength of the study, nor my previous comment on the initial description of such an approach. Something such as 'Colonisation pressure as a real-time machine-learning risk metric for nosocomial acquisition: a proof-of-concept using routinely collected EHR data' might be better suitable to make it more convincing and suitable.

(Remarks on code availability)

The code is quite clean. Explicit use of package names is a bit unnecessary (could just have used `library()` on top), but it does make all function calls unambiguous.

I noticed different style in the code, e.g. sometimes usage of `base R `order()``, sometimes usage of ``dplyr::arrange()``, and sometimes using `data.table` for data handling. Might be due to different authors working on the same code.

But most importantly, the overall structure and organisation of the code are more than satisfactory - if all research projects looked as structured as this, the scientific world would undoubtedly be a better place!

Reviewer #3

(Remarks to the Author)

This is a well-conducted and interesting study on colonization pressure (CP) and hospital-acquired infections, covering a broad set of organisms across multiple hospitals. The work is novel in extending CP research beyond ICUs and resistant organisms, and the release of analytic code and dataset is a major strength. The associations you report between CP and

acquisition, including inverse associations for some non-cognate organisms, are noteworthy. The methodology is generally rigorous, but there are a few areas where clarification or further work would strengthen the manuscript.

Major comments:

Ward-level matching and infection prevention and control (IPC) confounding

CP is defined strictly as the prevalence of colonized patients in a ward. It represents the reservoir of potential exposure. IPC measures such as hand hygiene, environmental cleaning, staffing ratios, and room design are not components of CP itself but rather act as modifiers of transmission once CP exists. In other words, high CP creates greater exposure potential, while IPC measures determine whether that exposure translates into actual acquisition.

In this study, cases and controls were not matched within the same ward. This raises the possibility that some of the observed associations between CP and acquisition reflect unmeasured ward-level differences in IPC performance or layout, rather than CP alone. For example, two wards with similar CP but differing hand hygiene compliance could exhibit different acquisition risks. Without accounting for ward-level clustering, it is difficult to fully separate the effect of CP from the modifying role of ward-specific IPC factors.

While the authors do acknowledge the role of IPC (e.g., contact precautions, room occupancy, healthcare worker vectors) in their discussion, this point may warrant further statistical exploration. For instance, sensitivity analyses that restrict cases and controls to the same ward, or inclusion of ward identity as a clustering/random effect, would help clarify the extent to which the observed CP–acquisition associations are independent of ward-level IPC practices.

Minor points

1. The models had poor AUROC and predictive value. You do note this, but the current framing could be misread as an attempt at patient-level prediction. I would suggest reframing the analysis as proof-of-principle, showing that CP is a useful ecological marker but not sufficient for individual prediction.
2. Ward design and room occupancy were not included in the analysis. The discussion mentions double occupancy rooms, but no systematic data were presented. Given that colonization pressure likely operates differently in settings with predominantly single rooms compared to cohort wards or multi-bedded bays, the generalizability of the findings to health systems with shared occupancy should be acknowledged more explicitly.
3. Figures 4–5 are dense and not easy to follow; simplifying them or improving captions would help readers.
4. The discussion of inverse associations as ecological competition should be more cautious, framing this as hypothesis-generating rather than an established mechanism.

(Remarks on code availability)

Version 1:

Reviewer comments:

Reviewer #1

(Remarks to the Author)

I appreciate the authors' changes. Citations for the assumption around antibiotic duration to disrupt the microbiome would be appreciated.

(Remarks on code availability)

I appreciate the authors' changes. Citations for the assumption around antibiotic duration to disrupt the microbiome would be appreciated.

Reviewer #2

(Remarks to the Author)

In the previous round, I was listed as Reviewer #2 in the rebuttal document. I am happy to continue our academic discussion of your work, which I continue to appreciate.

I must say, I am very impressed by the revisions the authors have made. I had raised a number of serious concerns, which I will briefly summarise below (following the order of the rebuttal), along with my response to the authors' replies. I will indicate where I believe further action is warranted.

- Overall: line numbers:

Thank you for adding these, it makes communication much easier on this platform.

- Be more explicit about the limitations imposed by relying solely on routine diagnostic data / adding that the work is more a proof-of-concept:

To my (relief/great joy), the authors have made substantial efforts to clarify this point, both in the Discussion and in the title,

as I had suggested. I am pleased to see this agreement, and I consider this a significant improvement.

- Limits of generalisability / focus on US-only data

Given the clarification that this study should be regarded as a proof-of-concept, I consider this point adequately addressed. While a more explicit statement might further improve interpretation, I do not believe it is strictly necessary.

- XGBoost performance

The issue of over-interpretation has been well addressed. Thank you for making this clearer in the Discussion.

- Comparisons with prior work

Figure S6 is a very welcome addition, and I particularly appreciate the inclusion of lines L250-269 (according to the rebuttal that is, in the PDF they are L243-260, but this is fine). One small issue is that *C. difficile* was never written out with the full genus (i.e., *Clostridioides difficile*). This should be added in Cohort Definition where it is mentioned for the first time. Do keep in mind that it's *Clostridioides*, not *Clostridium* as it used to be until some years ago.

- PhysioNet, not able to locate the data set

Thank you for making the dataset publicly accessible and providing a functional DOI. As noted previously, this is excellent practice.

I would suggest making a stronger case for this openness by slightly rephrasing the opening sentence of the "Source code and data availability" section as follows: 'To promote transparency and reproducibility, we have made both the full source code for the analytic pipeline, from raw data pre-processing to model output, and the de-identified dataset publicly available on GitHub.'

This is something worth highlighting; show that commitment with pride :)

- Title rephrase

As noted, this point has been fully addressed.

Code:

Thank you for the thoughtful discussion and engagement with my review. I have nothing to add here. Excellent work.

As the authors will note, only a few very minor suggestions remain. Following these final revisions, I would encourage the editor to give strong consideration to accepting the manuscript for publication.

(Remarks on code availability)

Reviewer #3

(Remarks to the Author)

(Remarks on code availability)

REVIEWER COMMENTS

Reviewer #1 (Remarks to the Author):

This is an interesting paper evaluating the colonization pressure for multiple pathogens in a healthcare system. Unfortunately, I am not able to access the supplementary materials so I cannot fully review this manuscript.

We thank the reviewer for taking the time to read our manuscript and apologize for not being able to access the supplemental material and code base. The revised manuscript has all of the supplemental material and the code base is available at https://github.com/sanjatkanjilal/hospital_onset_infection_analyses.

This sentence is confusing to me, “Controls were also required to have a length of stay in their room that was within ± 3 days of a case's length of stay, or the date timestamp of collection for the case's index culture, whichever was shorter.” Could an example be given?

We have added an example to clarify the noted sentence: “For example, if a case was in a room for 20 days and the culture with the target pathogen was identified on day 12, the matched control was in their room between 9 to 15 days”.

Is there a differential bias of this inclusion criterion, “stayed in only one room for >48 hours during the entirety of the observation period”? How did this impact the data included in the analysis?

The reviewer brings up a good point about bias being introduced by our strict inclusion criteria. We deliberately required cases and controls to stay in a single room for 48 hours as we felt the benefit of eliminating confounding introduced by exposure to a different ward's microbiome was greater than the risk of selection bias. However, we acknowledge that our patient cohorts may not be representative of all hospitalized patients, particularly ICU patients, who are often discharged to a general ward after stabilization. We have modified the title of this article to state this is a ‘proof-of-concept’ study and have expanded the limitations section in the discussion accordingly. We expect our patients on average to be less medically complicated than patients who require transfer between wards, with some non-specificity attributable to patients who move between wards for non-medical reasons (ie to balance hospital capacity).

Old text	Updated text
Lines 346 - 348: “Furthermore, restricting the observation period to the first room prevents confounding that may occur if the patient was exposed to another hospital unit during the same hospitalization. “	Lines 390 - 395: “Restricting the analysis to patients with a single room stay prevents confounding that may occur if the patient was exposed to another hospital unit during the same hospitalization, but leads to reduced generalizability as patients often transfer between wards, such as those admitted to the ICU. As noted earlier, our results represent a proof-of-principle that CP impacts nosocomial acquisition, but we cannot extrapolate its importance relative to other factors with this study.”

On page 5, is it “never had the organism” or “never had the organism identified”?

We thank the reviewer for noticing the wording of this sentence. We have updated it to say ‘identified’.

Could the authors comment on the impact of their definition of a course of antibiotics, “A course was defined as >2 days of treatment >7 days apart and was stratified by the following antibiotic classes”? What proportion of patients received 2 or fewer days of antibiotics? Is the misclassification differential?

We acknowledge the reviewer’s concern that this choice may further reduce generalizability. The decision to exclude antibiotic courses <2 days in length was made for two reasons: 1) to help filter out spurious prescriptions (ordered but not administered), and 2) it reduces the number of antibiotic courses we needed to account for in matching cases to controls, thereby increasing our sample size. Implicit in this decision is the assumption that at least 2 days of systemic antibiotic exposure are necessary to make a lasting impact on a host’s gut and skin microbiome. While several studies have examined antibiotic duration and microbiome perturbation, the majority have examined courses >3 days in length which make results difficult to generalize. Nonetheless, the literature suggests at least 72 hours of exposure are needed.

Unfortunately, our filtering criteria make it difficult to know what proportion of antibiotic courses were <48 hours in duration by each target pathogen cohort. In the overall antibiotic course dataset (which includes a row for every course given to any patient at our hospital since 2015), 45% of courses were for <2 days in duration. The distribution of the top 10 antibiotics is shown below, and account for 57% of all prescriptions. The table indicates the majority of these are for outpatients and are narrow spectrum. Additionally, since these are likely for outpatients, there is no way of confirming whether the patient actually took them as prescribed; for instance, a doxycycline prescription for <2 days in duration suggests a ‘pill-in-pocket’ for post-exposure prophylaxis with Lyme disease that can be taken at any time. Nitrofurantoin courses are typically

5 days, therefore a course for <2 days may represent either a step-down regimen from a more broad antibiotic (which would be matched upon in our study) or a spurious (or cancelled) prescription. For these reasons, we expect these courses to have a relatively limited impact on the microbiome.

Top 10 antibiotic courses with duration <2 days across Mass General Brigham (2015 - 2025)

antibiotic	count
DOXYCYCLINE	554076
AMOXICILLIN-CLAVULANATE	538904
AZITHROMYCIN	481506
AMOXICILLIN	468416
CEPHALEXIN	368068
TRIMETHOPRIM-SULFAMETHOXAZOLE	364366
NITROFURANTOIN	314655
CEFAZOLIN	301562
CIPROFLOXACIN	264111
CEFTRIAXONE	242554

Similarly, what was the impact of this inclusion criterion, “Other study inclusion criteria included not being actively treated for infection, defined as no record of antibiotic exposure from day -7 to day +3 relative to the time of entry into their room.”

Similar to the point earlier, the goal of our inclusion criteria was to select for a population that would allow us to isolate the effects of a hospital ward microbiome as cleanly as possible. Broadly, we selected for the following characteristics:

1. No evidence of prior carriage
2. No evidence of active infection
3. Similar antimicrobial exposures between cases and controls
4. No risk of exposure to environmental microbiomes from other wards in the 90 days before or the 30 days after admission

These criteria will lead to the selection of patients admitted for non-infectious issues (heart failure or COPD exacerbations, hyperglycemia, stroke, MI, arrhythmia, renal failure, hypertensive crisis, etc) that can be managed by a single clinical service. We expect them to be less medically complicated than patients who require transfer between services, and the ICU in particular. However, we note in the flow diagrams (now attached in the supplement) that the drop in sample size after requiring pathogens to be identified in their first room is modest compared to the drops in sample size that occurred after requiring no prior antibiotic treatment, and observation of the target pathogen from day +3 to day +30. The median length of stay for our cohorts ranged from 3.8 days (*E. coli* controls) to 7.3 days (VAN-R *E. faecium* cases), with wide interquartile ranges. While these patients may not be fully representative of the general hospitalized population, they do overlap with the most common indications for hospital admission in the US.

The authors state that they adhered to STORBE, however, they provide limited information on participants, in particular items 13a+b (copied below). As I cannot see the supplemental files I'm not sure what is included in supplemental table 2 but there are insufficient data in the main text to evaluate this.

(a) Report numbers of individuals at each stage of study—eg numbers potentially eligible, examined for eligibility, confirmed eligible, included in the study, completing follow-up, and analysed

(b) Give reasons for non-participation at each stage

(c) Consider use of a flow diagram

We thank the reviewer for mentioning the STROBE guidelines and apologize again for not having the supplement present in the last version. It has been included in this version of the submission and includes flow diagrams (in table form) for each of the 11 cohorts (Table S2).

Reviewer #1 (Remarks on code availability):

I am not able to access the supplemental files and therefore cannot fully vet the code.

We apologize for not being able to access the code base. It is now publicly available on github at https://github.com/sanjatkanjilal/hospital_onset_infection_analyses.

Is there cross validation happening in the conditional logistic regression? For the conditional logistic regression it doesn't appear that the data is subset at all. Are AUROCs reported for the LR in the supplement (which I cannot access)?

This is an error on our part and we thank the reviewer for pointing this out. Model performance was assessed using the entire dataset, not with cross validation. The text in the methods section has been updated. The performance (AUROCs) of the conditional logistic regression models are reported in the supplement.

Is the data being split 80/10/10 for the xgboost modeling? For xgboost the data only appears to be split 80/20 with the test set being used for both testing and validation. But this isn't where they are choosing the model hyperparameters so this code might not be shared/a version issue. Adding to that assumption, the output files from this script are being saved to another folder than the data is loaded from for plotting the final results.

We thank the reviewer for noticing this. The version of the github code repository was out of date and has since been updated to include the 80/10/10 split of the XGBoost models. The output file paths are also now updated with a general file path.

Reviewer #2 (Remarks to the Author):

Sagers et al have developed an open-source informatics tool to estimate hospital unit-level colonisation pressure from EHR data, and used it to model nosocomial acquisition risk across a broad range of both drug-susceptible and drug-resistant pathogens. I believe this study is notable for its methodological thoroughness, its inclusion of a wide spectrum of organisms, large number of included patients, and its transparent, reproducible design supported by public release of data and code. I have also reviewed the code.

I would have liked continuous line numbers to be better able to refer to parts of the manuscripts, but this might be a limitation of the Nature submission platform? In any case, these are my comments of which I believe would increase the quality and clearness of the paper once addressed.

We thank the reviewer for taking the time to read our manuscript and providing very helpful comments to improve its quality.

My background is in both clinical microbiology and data science. Therefore, I understand quite well how important efforts like the proposed approach are and I'm always happy to read about them in new submissions. Although:

- I feel the authors could have been considerably more explicit about the limitations imposed by relying solely on routine diagnostic data. While I understand and appreciate the rationale for using EHR-based surveillance, it would strengthen the discussion to more directly acknowledge the selective nature of clinical testing and its variation across hospitals and regions. In my view, it is important to recognise that while such modelling may complement existing surveillance infrastructure, it cannot substitute for active surveillance in terms of sensitivity or comprehensiveness.

The reviewer makes a great point regarding biases introduced by clinical testing as opposed to surveillance testing. We now acknowledge this in the limitations section of the discussion. We fully agree that even if modeling CP were to show utility in the general hospitalized population, this approach is meant to complement and not replace existing active surveillance efforts, which remain critical for breaking transmission chains.

Old text	Updated text
Lines 260 - 262: “Our findings reinforce the hypothesis that colonization pressure is a proxy of a unit’s microbiome, and open up new and potentially important avenues of research for hospital epidemiology and microbial ecology.”	Lines 288 - 290: “Our findings reinforce the hypothesis that colonization pressure is a proxy of a unit’s microbiome and open up new avenues of research for microbial ecology and hospital epidemiology with the potential to augment existing infection control practices.”
	Lines 373 - 375: “In its current form, our tool and models represent a proof-of-concept and require further refinement before they are ready to augment existing infection control interventions.”
Lines 350 - 358: “Another limitation is our inability to identify silent carriage of a target organism, which could confound the relationship between CP and hospital acquisition. This stems partly from the fact that we have a lack of insight into events (microbiology specimens, antibiotic prescriptions) that occur outside of our healthcare system. We have reduced this possibility by excluding patients with evidence of the target organism in the previous 6 or 12 months. Finally, our data come from a single healthcare system in the northeast US and caution is warranted when extrapolating to other geographic areas and healthcare systems that may differ in terms of patient demographics, antibiotic consumption and infection control practices.”	Lines 396 - 407: “Another limitation is our inability to identify silent carriage of a target organism, which could confound the relationship between CP and hospital acquisition. This stems partly from the fact that many of our target organisms do not have surveillance programs for asymptomatic sampling, thus our insight into the true burden of disease is dependent on recognition by providers, which is highly variable. Additionally, we lack insight into events (microbiology specimens, antibiotic prescriptions) that occur outside of our healthcare system. We have reduced this possibility by excluding patients with evidence of the target organism in the previous 6 or 12 months and our antibiotic exposure matching strategy. Finally, our data come from a single healthcare system in the northeast US and caution is warranted when extrapolating to other geographic areas and healthcare systems that may differ in terms of patient demographics, clinical sampling practices, antibiotic consumption, and infection control policies.”

- Given that all data were derived from a single integrated healthcare system in the northeastern United States, I feel the authors could be more forthright about the limits of generalisability. While

the release of code and data is commendable and promotes reproducibility, institutional variation in diagnostic practices, antimicrobial stewardship, and infection control protocols could significantly impact both colonisation pressure and acquisition risk. This ought to be acknowledged more prominently, since it cannot be known how effective this model/approach would be in other sites. In other words, the study lacks external validation. Given the journal quality, I would propose that the authors either (1) add a complete external validation using another hospital site, which I understand is probably unfeasible, or (2) add to the manuscript that the paper describes a proof-of-concept rather than a validated approach. This should at least be clear in the title and/or abstract.

The reviewer is correct that we lack external validity for this approach and that this is a proof-of-concept study. Given the highly curated and heavily pre-processed nature of our final dataset, an external validation is unfortunately not feasible within a reasonable time frame. However, it is our hope that other research groups use our heavily annotated code base and de-identified data to reproduce, and improve upon, our analysis and thereby support external validation in a ‘federated’ manner. We have adjusted the title of the manuscript per their suggestion and modified the discussion to highlight that further work / research is necessary to make this ready for use by infection control.

- Reading the section on XGBoost performance (which, I must admit, was somewhat disappointing - didn't expect a great decision tree model such as XGBoost to perform that bad), I found the relatively poor discriminatory capacity somewhat downplayed. While I agree that CP may be only one of many necessary features for accurate prediction, the limitations of the ML approach in its current form should be more clearly acknowledged. In particular, I would suggest the authors guard against overinterpreting the current utility of these models for real-time risk stratification.

The performance of the XGBoost models were also disappointing to us, though within range of prior analyses we have performed using EHR data to predict AMR. We appreciate the need to be cautious about over-interpretation and have modified the discussion and limitations section accordingly.

Old text	Updated text
Lines 326 - 331: "It is important to note however, that despite the advantages of using a machine learning approach to maximize predictive power, our XGBoost models had low positive and negative predictive value for hospital acquisition. This may be explained by small sample sizes and the fact that colonization pressure is one of many factors	Lines 369 - 375: "It is important to note however, that despite the advantages of using a machine learning approach to maximize predictive power, our XGBoost models had low positive and negative predictive value for hospital acquisition. This may be explained by small sample sizes and the fact that colonization pressure is one of many factors that must be included in real-time models to support decision-making. In its current form,

that must be included in real-time models to support decision-making.”

our tool and models represent a proof-of-concept and require further refinement before they are ready to augment existing infection control interventions.”

- The paper makes a strong case for expanding the concept of colonisation pressure to include drug-susceptible organisms and non-ICU wards. Still, I would have appreciated a more direct comparison with prior work, especially in terms of quantitative effect sizes. In my view, situating this study more firmly within the context of prior colonisation pressure studies would clarify what is genuinely novel versus what confirms established findings in a broader setting.

The reviewer makes a good point about more explicitly situating our findings in the context of prior work. To aid in this we have undertaken additional analyses to provide more interpretable estimates of the impact of colonization pressure:

1. We modeled the number of people (i.e., ward co-occupants) necessary to generate a unit increase in colonization pressure, stratified by a range of median days since prior infection (30, 60, 90, 120, 150 and 180 days). This is shown in figure S6 in the supplement.
2. We then estimated the number of people needed to generate the colonization pressures for each pathogen cohort, differentiating between cases and controls and stratifying by median days since prior infection (figures S7 - S11).

We refer to these figures in new paragraphs in the results section and the discussion. The latter compares our results to the existing literature using the values from the new analyses. We have endeavored to highlight how our study supports previous work while also exploring new territory.

Updated text:

Lines 250 - 269: **”Interpretable effects of colonization pressure**

To better understand the impact of colonization pressure in concrete terms, we estimated the number of ward co-occupants associated with a unit change in CP (supplementary figure S6). Assuming median times since infection of 30, 90 and 180 days, a 1 unit increase in CP requires 1.4, 2.3 and 3.6 ward co-occupants to have had the CP organism, respectively. We then estimated the number of ward co-occupants with prior acquisition from each of the 9 CP organism sets, for each of the 11 target pathogen cohorts, differentiating between cases and controls, and stratifying by median time to infection (supplemental figures S7 through S11). As an example, the greatest positive effect size was for CP_{C. difficile} and hospital acquisition of *C. difficile* (+32.5%, 95% CI +21.9% to +44.0%). Given the median CP_{C. difficile} of 1.46 for the cohort of *C. difficile* cases, 2.1, 3.4 and 5.3 ward co-occupants would be estimated to have had *C. difficile*, a median of 30, 90 and 180 days prior to the entry of cases into the ward, respectively, compared to 1.2, 2.0 and 3.2 ward co-occupants in the control group. Similarly, the greatest negative effect size was for CP_{PSA-DR} and vancomycin susceptible *E. faecalis* (-10.0%, 95% CI -15.6% to -4.0%). In the cohort of

cases who acquired vancomycin susceptible *E. faecalis*, the median CP_{PSA-DR} was 0.69, which translates to 1.0, 1.6 and 2.6 ward co-occupants with drug resistant *P. aeruginosa*, a median of 30, 90 and 180 days prior to the entry of cases into the ward, respectively, compared to 1.1, 1.7 and 2.8 ward co-occupants in the control group. These values provide a point of reference for the impact of CP on target pathogens in terms of numbers of patients.”

Lines 302 - 321: “While it is difficult to directly compare our study to others due to differences in methods, our results are largely in line with prior research examining colonization pressure. Lawrence et al defined *C. difficile* ICU CP as the sum of a daily point prevalence, and found a statistically significant rate of acquisition when CP crossed a threshold of 10 patient-days of exposure³¹, which is comparable to our estimate of 2 to 5 ward co-occupants with prior infection between 30 and 180 days in the past (supplementary figure S8). Similarly, Jolivet et al defined CP as the ratio of colonized ICU patient-days to total ICU patient-days and found a threshold of 100 to 200 patient days to be associated with odds of acquisition for ESBL Enterobacteriales¹², which is an order of magnitude weaker than for *C. difficile*. This is within range of our estimates that show ESBL *K. pneumoniae* acquisition to be associated with a CP of between 4 and 12 ward co-occupants having ESBL Enterobacteriales 30 to 180 days prior (supplementary figure S7). Finally, Chanderraj et al used a matched case control design to examine the influence of VRE CP on VRE acquisition in the ICU, in addition to patient-level factors such as antibiotics and proton pump inhibitor exposure, and found no association¹⁵. This is in agreement with our findings that VRE acquisition was not associated with CP from VRE, though it was associated with CP from VSE. Importantly, all of these studies were performed in ICUs, which are mostly single room occupancy, and none performed matching to equalize prior antibiotic exposures. To our knowledge, no studies have examined relationships between CP and drug susceptible organisms or non-cognate pairs, and no studies have found inverse associations in multivariate models.”

- I was pleased to see that the authors have decided to make both their codebase and de-identified dataset publicly available. In my view, this represents a strong commitment to transparency and reproducibility, and sets an example that others in the field would do well to follow. However, at PhysioNet (<https://physionet.org>), I could not find the project entitled ‘Predictors of hospital onset infection: A matched retrospective cohort dataset’ as the authors indicate, or when searching for parts of that or for the manuscript title. Importantly, PhysioNet provides DOIs - please explicitly provide it in the manuscript to prevent others for searching in vain.

We thank the reviewer for their approval of our effort to support open science. The dataset was submitted to Physionet on July 14th and we were awaiting their editors' decision to make it public while this submission was being reviewed. It is now publicly available at <https://doi.org/10.13026/pa4z-5293>, and we have updated the manuscript with the live link. We will eventually publish a separate paper in a data journal describing this dataset in detail, as it contains information for a complementary analysis on hospital acquisition that we have not yet completed.

Lastly, I feel the title does not encompass the strength of the study, nor my previous comment on the initial description of such an approach. Something such as 'Colonisation pressure as a real-

time machine-learning risk metric for nosocomial acquisition: a proof-of-concept using routinely collected EHR data' might be better suitable to make it more convincing and suitable.

We appreciate this suggestion and have updated the title of the manuscript!

Reviewer #2 (Remarks on code availability):

The code is quite clean. Explicit use of package names is a bit unnecessary (could just have used ``library()`` on top), but it does make all function calls unambiguous.

Thank you for reviewing our code base. We acknowledge the explicit use of package names with every command can be a bit unwieldy but decided to use this given issues we've had with package conflicts.

I noticed different style in the code, e.g. sometimes usage of base R ``order()``, sometimes usage of ``dplyr::arrange()``, and sometimes using `data.table` for data handling. Might be due to different authors working on the same code.

The reviewer is correct that the code base reflects the work of multiple authors over three years. We selected `data.table` for several functions as it was 10-100X faster in processing very large datasets than the tidyverse libraries.

But most importantly, the overall structure and organisation of the code are more than satisfactory - if all research projects looked as structured as this, the scientific world would undoubtedly be a better place!

We deeply appreciate this comment. A great deal of effort was put into making the code usable and the data publicly available! We sincerely hope other groups can use it and improve upon it.

Reviewer #3 (Remarks to the Author):

This is a well-conducted and interesting study on colonization pressure (CP) and hospital-acquired infections, covering a broad set of organisms across multiple hospitals. The work is novel in extending CP research beyond ICUs and resistant organisms, and the release of analytic code and dataset is a major strength. The associations you report between CP and acquisition, including inverse associations for some non-cognate organisms, are noteworthy. The methodology is generally rigorous, but there are a few areas where clarification or further work would strengthen the manuscript.

Major comments:

Ward-level matching and infection prevention and control (IPC) confounding

CP is defined strictly as the prevalence of colonized patients in a ward. It represents the reservoir of potential exposure. IPC measures such as hand hygiene, environmental cleaning, staffing ratios, and room design are not components of CP itself but rather act as modifiers of transmission

once CP exists. In other words, high CP creates greater exposure potential, while IPC measures determine whether that exposure translates into actual acquisition.

In this study, cases and controls were not matched within the same ward. This raises the possibility that some of the observed associations between CP and acquisition reflect unmeasured ward-level differences in IPC performance or layout, rather than CP alone. For example, two wards with similar CP but differing hand hygiene compliance could exhibit different acquisition risks. Without accounting for ward-level clustering, it is difficult to fully separate the effect of CP from the modifying role of ward-specific IPC factors.

While the authors do acknowledge the role of IPC (e.g., contact precautions, room occupancy, healthcare worker vectors) in their discussion, this point may warrant further statistical exploration. For instance, sensitivity analyses that restrict cases and controls to the same ward, or inclusion of ward identity as a clustering/random effect, would help clarify the extent to which the observed CP–acquisition associations are independent of ward-level IPC practices.

The reviewer brings up an excellent point regarding the impact of IPC measures as a mediator of exposure from the ward microbiome and we agree accounting for them is important for interpreting the results of our models. As this study was designed to examine the impact of ward level CP on hospital acquisition, matching on that variable is not possible. As suggested, we investigated the possibility of incorporating hospital unit as a random effect, but found that it is not compatible with conditional logistic regression, which is the appropriate analysis method for our matched case-control design.

Conditional logistic regression works by conditioning on the matched sets, which mathematically eliminates the matched-set-specific intercepts (nuisance parameters) from the likelihood function. This conditioning process is essential for properly accounting for the matching on prior antibiotic exposure and avoiding bias in our effect estimates. Adding random effects for hospital ward would require a different modeling framework (generalized linear mixed models) that estimates random effect distributions through integration over the likelihood, which is incompatible with the conditional likelihood approach used in matched case-control analysis.

The statistical literature confirms that matched case-control data is best analyzed using conditional logistic regression that stratifies by the matching variables^{1,2}. We could not identify any textbook or research publications that combined this with random effects for additional clustering within the same model framework.

We next tried to add the ward ID as a fixed effect to our models but given the very large number of wards in each cohort, the model failed to converge on a coefficient estimate, therefore ultimately we left it out of the analysis. Unfortunately, given ambiguities in the source data, it was not possible to collapse ward ID into categories. The published dataset does include ward ID (named 'dept_code') for further analysis by outside investigators.

References:

1. Breslow NE, Day NE. Statistical methods in cancer research. Volume I - The analysis of case-control studies. IARC Sci Publ. 1980;(32):5-338. PMID: 7216345.

2. Hosmer, D. W., Lemeshow, S. & Sturdivant, R. X. Applied Logistic Regression. Chapter 7: Logistic Regression for Matched Case- Control Studies Wiley Ser. Probab. Stat. 243–268 (2016) doi:10.1002/9781118548387.ch7.

Minor points

1. The models had poor AUROC and predictive value. You do note this, but the current framing could be misread as an attempt at patient-level prediction. I would suggest reframing the analysis as proof-of-principle, showing that CP is a useful ecological marker but not sufficient for individual prediction.

This is duly noted and we have adjusted the title of the manuscript to state this is a proof-of-concept study. We have also updated the discussion section to be more cautious in the interpretation of our results.

Old text	Updated text
Lines 326 - 331: "It is important to note however, that despite the advantages of using a machine learning approach to maximize predictive power, our XGBoost models had low positive and negative predictive value for hospital acquisition. This may be explained by small sample sizes and the fact that colonization pressure is one of many factors that must be included in real-time models to support decision-making."	Lines 369 - 375: "It is important to note however, that despite the advantages of using a machine learning approach to maximize predictive power, our XGBoost models had low positive and negative predictive value for hospital acquisition. This may be explained by small sample sizes and the fact that colonization pressure is one of many factors that must be included in real-time models to support decision-making. In its current form, our tool and models represent a proof-of-concept and require further refinement before they are ready to augment existing infection control interventions."

2. Ward design and room occupancy were not included in the analysis. The discussion mentions double occupancy rooms, but no systematic data were presented. Given that colonization pressure likely operates differently in settings with predominantly single rooms compared to cohort wards or multi-bedded bays, the generalizability of the findings to health systems with shared occupancy should be acknowledged more explicitly.

The reviewer raises an important point. Unfortunately we did not have access to ward occupancy types in our clinical data warehouse, and they are constantly in flux over time. The data comparing single vs double occupancy rooms is mixed but overall suggests single occupancy is associated with lower rates of hospital-acquired infection. Presumably single-occupant wards would have lower colonization pressure as they would have fewer patients as reservoirs for transmission to healthcare workers and surfaces. However, some single-occupant wards are ICUs, where the

burden of colonization is often higher due to the acuity and comorbidity of patients requiring this level of care. As our hospital system is a mix of single and multi-occupant rooms, we agree with the reviewer that the generalizability to hospitals that are solely single occupant rooms may be limited, but the results may be typical of other mixed hospitals. We have updated the text to reflect this insight.

Old text	Updated text
Lines 355 - 359: "Finally, our data come from a single healthcare system in the northeast US and caution is warranted when extrapolating to other geographic areas and healthcare systems that may differ in terms of patient demographics, antibiotic consumption and infection control practices. This could be surmounted by combining data across multiple health systems using a common data model or a federated learning approach."	Lines 403 - 410: "Finally, our data come from a single healthcare system in the northeast US and caution is warranted when extrapolating to other geographic areas and healthcare systems that may differ in terms of patient demographics, clinical sampling practices, antibiotic consumption, and infection control policies. In particular, as our hospital contains a mix of single and multi-occupant rooms, these results may not generalize to newer hospitals that contain solely single occupant rooms. This could be surmounted by including occupancy data and combining data across multiple health systems using a common data model or a federated learning approach."

3. Figures 4–5 are dense and not easy to follow; simplifying them or improving captions would help readers.

We thank the reviewer for this suggestion and have modified the figures and legends to be more interpretable and consistent with terminology in the main manuscript.

Old legends	Updated legends
Figure 4: Impact of colonization pressure on adjusted odds of hospital acquisition of cognate organisms. A) Impact of CP from enteric organisms on hospital acquisition of an enteric organism. B) Impact of CP from skin flora on hospital acquisition of skin flora. C) Impact of CP from environmental flora on hospital acquisition of environmental flora.	Figure 4: Impact of colonization pressure on odds of nosocomial acquisition of target cognate organisms. A) Impact of CP from enteric organisms on hospital acquisition of target enteric organisms. B) Impact of CP from skin flora on hospital acquisition of target skin flora. C) Impact of CP from environmental flora on hospital acquisition of target environmental flora.

Figure 5: Impact of colonization pressure on adjusted odds of hospital acquisition of non-cognate organisms. A) Impact of CP from enteric organisms on hospital acquisition of skin and environmental organisms. B) Impact of CP from skin flora on hospital acquisition of enteric and environmental organisms. C) Impact of CP from environmental flora on hospital acquisition of enteric and skin organisms.

Figure 5: Impact of colonization pressure on odds of nosocomial acquisition of target non-cognate organisms. A) Impact of CP from enteric organisms on hospital acquisition of target skin and environmental organisms. B) Impact of CP from skin flora on hospital acquisition of target enteric and environmental organisms. C) Impact of CP from environmental flora on hospital acquisition of target enteric and skin organisms.

Old figures

Figure 4

New figures

Figure 4

Figure 5

Figure 5

4. The discussion of inverse associations as ecological competition should be more cautious, framing this as hypothesis-generating rather than an established mechanism.

We have modified the discussion of inverse associations to be more cautious in our interpretations.

Old text	Updated text
Lines 309 - 318: "A third possible explanation for the inverse relationship is that contamination of a ward by one organism may prevent other organisms from establishing themselves. This may be due to repeated contamination by colonized hosts on a given floor, or alternatively, intra- and inter-species competition on hospital environmental surfaces. The concept of colonization resistance and competition has been described in vitro and for niches across the human body³⁵⁻³⁸. Further research is necessary to prove whether founder effects and other competitive mechanisms take place on high touch surfaces, sinks and bathrooms within hospital units. A causal relationship offers the possibility of leveraging competition between organisms to reduce rates of transmission and hospital-acquired infection from difficult to treat pathogens³⁹. On a more pragmatic level, it increases the utility of passive surveillance of unit-level CP."	Lines 353 - 360: "A third possible explanation for the inverse relationship is that contamination of a ward by one organism may prevent other organisms from establishing themselves. This may be due to repeated contamination by colonized hosts on a given floor, or alternatively, intra- and inter-species competition on hospital environmental surfaces. The concept of colonization resistance and competition has been described in vitro and for niches across the human body³⁴⁻³⁷, but further research is necessary to prove whether founder effects and other competitive mechanisms take place on high touch surfaces, sinks, and bathrooms within hospital units."

Sincerely,

Sanjat Kanjilal, MD, MPH

Group Leader | Microbiology Informatics Technology Lab

Dept of Medical Microbiology and Infection Prevention | Amsterdam UMC

s.kanjilal@amsterdamumc.nl

We appreciate the comments by our reviewers to our updated manuscript, now entitled “Using electronic health records to assess the relationship between colonization pressure and nosocomial pathogen acquisition” being considered for publication in *Nature Communications*.

Below you will find a point-by-point response to reviewer comments. Author responses are highlighted in red.

REVIEWER COMMENTS

Reviewer #1 (Remarks to the Author):

I appreciate the authors' changes. Citations for the assumption around antibiotic duration to disrupt the microbiome would be appreciated.

Below you will find a number of references related to the impact of antibiotic duration on microbiome perturbations. As noted in our previous response, the shortest duration of exposure studied was at least 72 hours in length, and they are mostly for broad spectrum antibiotics. Furthermore they study populations that differ substantially from those in our study, which makes results challenging to extrapolate. Overall, we felt it reasonable to exclude courses 2 days in length or shorter as they were mostly narrow-spectrum therapies and to filter out spurious prescriptions, post-exposure prophylactic courses, and step-down therapies.

1. d'Humières, C. *et al.* Perturbation and resilience of the gut microbiome up to 3 months after β -lactams exposure in healthy volunteers suggest an important role of microbial β -lactamases. *Microbiome* 12, 50 (2024).

- **Duration tested:** Cefotaxime or ceftriaxone for 3 days in healthy volunteers
- **Relevance:** The study collected 12 fecal samples over 90 days and found perturbations to bacterial microbiota, phage microbiota, and resistome composition.

2. Palleja, A. *et al.* Recovery of gut microbiota of healthy adults following antibiotic exposure. *Nat. Microbiol.* 3, 1255–1265 (2018).

- **Duration tested:** 4-day intervention with meropenem, gentamicin, and vancomycin in 12 healthy men
- **Relevance:** Demonstrated immediate blooms of pathobionts (*Enterobacteria*, *Enterococcus faecalis*, *Fusobacterium nucleatum*) and positive selection for species harboring β -lactam resistance genes. Nine common species remained undetectable even at 180 days post-treatment.

3. Pettigrew, M. M. *et al.* Comparison of the Respiratory Resistomes and Microbiota in Children Receiving Short versus Standard Course Treatment for Community-Acquired Pneumonia. *mBio* 13, e00195-22 (2022).

- **Duration tested:** Compared 5-day vs 10-day beta-lactam courses in children with pneumonia
- **Relevance:** Found that the shorter 5-day course increased antibiotic resistance genes in both respiratory and gut microbiomes, though to a lesser extent than 10-day courses.

Reviewer #2 (Remarks to the Author):

In the previous round, I was listed as Reviewer #2 in the rebuttal document. I am happy to continue our academic discussion of your work, which I continue to appreciate.

I must say, I am very impressed by the revisions the authors have made. I had raised a number of serious concerns, which I will briefly summarise below (following the order of the rebuttal), along with my response to the authors' replies. I will indicate where I believe further action is warranted.

We thank the reviewer for their very kind comments and would like to share our gratitude for their close reading of our manuscript. It has greatly improved the work.

- Overall: line numbers:

Thank you for adding these, it makes communication much easier on this platform.

- Be more explicit about the limitations imposed by relying solely on routine diagnostic data / adding that the work is more a proof-of-concept:

To my (relief/great joy), the authors have made substantial efforts to clarify this point, both in the Discussion and in the title, as I had suggested. I am pleased to see this agreement, and I consider this a significant improvement.

- Limits of generalisability / focus on US-only data

Given the clarification that this study should be regarded as a proof-of-concept, I consider this point adequately addressed. While a more explicit statement might further improve interpretation, I do not believe it is strictly necessary.

- XGBoost performance

The issue of over-interpretation has been well addressed. Thank you for making this clearer in the Discussion.

- Comparisons with prior work

Figure S6 is a very welcome addition, and I particularly appreciate the inclusion of lines L250-269 (according to the rebuttal that is, in the PDF they are L243-260, but this is fine). One small issue is that *C. difficile* was never written out with the full genus (i.e., *Clostridioides difficile*). This should be added in Cohort Definition where it is mentioned for the first time. Do keep in mind that it's *Clostridioides*, not *Clostridium* as it used to be until some years ago.

Thank you! As there is space in the manuscript, and given its importance for interpreting our results, we have moved Figure S6 into the main manuscript (now Figure 6). We have also now written out *Clostridioides difficile* at first mention in the results (which appears before methods in the final manuscript; line 92).

- PhysioNet, not able to locate the data set

Thank you for making the dataset publicly accessible and providing a functional DOI. As noted previously, this is excellent practice.

I would suggest making a stronger case for this openness by slightly rephrasing the opening sentence of the "Source code and data availability" section as follows: 'To promote transparency and reproducibility, we have made both the full source code for the analytic pipeline, from raw data pre-processing to model output, and the de-identified dataset publicly available on GitHub.' This is something worth highlighting; show that commitment with pride :)

We appreciate the suggestion and have updated the data and code availability statements based on journal requirements and the above recommendation.

Old (lines 169 - 173) Source code and data availability The source code for the entire analytic pipeline, spanning pre-processing of raw data to model output, is available on Github²¹. A de-identified version of the final cleaned dataset used for the CP models is available on Physionet^{22,23} (project title 'Predictors of hospital onset infection: A matched retrospective cohort dataset').	New (lines 404 - 414) DATA AVAILABILITY To promote transparency and reproducibility, we have made a de-identified version of the final cleaned patient-level data available under restricted access for the protection of human subjects and to ensure compliance with relevant data privacy regulations at Physionet^{22,23}. Access can be obtained by registering for a PhysioNet account, completing the required CITI Program training in human subjects research (Data or Specimens Only Research), and signing a Data Use Agreement (DUA). The raw patient-
---	--

	level data are protected and are not available due to data privacy laws. CODE AVAILABILITY The source code for the entire analytic pipeline, spanning pre-processing of raw data to model output, is available on Github²¹.
--	--

- Title rephrase

As noted, this point has been fully addressed.

Per journal requirements and editor suggestion, the title has been modified to: "Using electronic health records to assess the relationship between colonization pressure and nosocomial pathogen acquisition". It is mentioned throughout the manuscript (abstract and several times in the discussion) that this is a proof-of-concept study.

Code:

Thank you for the thoughtful discussion and engagement with my review. I have nothing to add here. Excellent work.

As the authors will note, only a few very minor suggestions remain. Following these final revisions, I would encourage the editor to give strong consideration to accepting the manuscript for publication.

Many thanks again!

Sincerely,

Sanjat Kanjilal, MD, MPH
Group Leader | Microbiology Informatics Technology Lab
Dept of Medical Microbiology and Infection Prevention | Amsterdam UMC
s.kanjilal@amsterdamumc.nl